# Artificial Intelligence Applications in Breast Imaging: Current Status and Future Directions

**DOI:** 10.3390/diagnostics13122041

**Published:** 2023-06-13

**Authors:** Clayton R. Taylor, Natasha Monga, Candise Johnson, Jeffrey R. Hawley, Mitva Patel

**Affiliations:** Department of Radiology, The Ohio State University Wexner Medical Center, Columbus, OH 43210, USA; natasha.monga@osumc.edu (N.M.); candise.johnson@osumc.edu (C.J.); jeffrey.hawley@osumc.edu (J.R.H.); mitva.patel@osumc.edu (M.P.)

**Keywords:** breast imaging, artificial intelligence, deep learning, machine learning, mammography, breast MRI, breast ultrasound, radiology workflow, computer-aided diagnosis, computer-aided detection

## Abstract

Attempts to use computers to aid in the detection of breast malignancies date back more than 20 years. Despite significant interest and investment, this has historically led to minimal or no significant improvement in performance and outcomes with traditional computer-aided detection. However, recent advances in artificial intelligence and machine learning are now starting to deliver on the promise of improved performance. There are at present more than 20 FDA-approved AI applications for breast imaging, but adoption and utilization are widely variable and low overall. Breast imaging is unique and has aspects that create both opportunities and challenges for AI development and implementation. Breast cancer screening programs worldwide rely on screening mammography to reduce the morbidity and mortality of breast cancer, and many of the most exciting research projects and available AI applications focus on cancer detection for mammography. There are, however, multiple additional potential applications for AI in breast imaging, including decision support, risk assessment, breast density quantitation, workflow and triage, quality evaluation, response to neoadjuvant chemotherapy assessment, and image enhancement. In this review the current status, availability, and future directions of investigation of these applications are discussed, as well as the opportunities and barriers to more widespread utilization.

## 1. Introduction

Breast cancer is the most common cancer in women of the United States, excluding skin cancers, and represents nearly 1 in 3 new female cancers each year. According to the American Cancer Society, there will be nearly 300,000 new cases of invasive breast cancer and over 50,000 cases of ductal carcinoma in situ diagnosed in 2023, with over 43,000 deaths attributable to breast cancer in the United States alone [1]. The high incidence and burden of breast cancer represent a tremendous challenge and opportunity for breast cancer screening programs. The purpose of any breast cancer screening program is to reduce the morbidity and mortality of breast cancer by identifying early, small breast cancers to ensure accurate diagnosis and optimal treatment. Screening mammography is the only breast cancer screening modality with a proven mortality benefit, leading to the widespread adoption of mammography-based screening programs throughout the world.

Population based screening efforts have led to a large number of mammograms being performed annually, with nearly 40 million mammograms performed every year in the United State alone [2]. The importance of screening mammography performance to breast cancer screening programs and the sheer volume of mammograms involved create an imperative need to maximize performance and quality. In the United States, this is closely regulated by the Food and Drug Administration (FDA) via the Mammography Quality Standards Act (MQSA), including recent emphasis via the Enhancing Quality Using the Inspection Program (EQUIP) process initiated in 2017. These processes have helped ensure quality and uniformity among screening mammograms performed in the United States. However, even with these efforts there remain opportunities for improvement in performance metrics for screening mammography. As an illustration of this need, an evaluation of performance by the Breast Cancer Surveillance Consortium found a sensitivity of 86.9% and a specificity of 88.9% for screening mammography, with opportunities for improvement particularly noted regarding abnormal interpretation rates (false positives) in nearly half of the studied radiologists [3].

## 2. Background

The convergence of screening mammography as a widespread population health tool with opportunities to improve performance to reduce breast cancer mortality has generated significant interest and research. Using computers in an attempt to improve performance is not new and has a long history in breast imaging in the form of computer-aided detection (CAD). The FDA first approved CAD for use in mammography in 1998, and by 2002, this technology was reimbursable by the Center for Medicare and Medicaid Services. This approval led to its rapid adoption in breast imaging, with 74% of mammograms in 2008 performed with CAD [4].

The initial excitement and enthusiasm for the benefits of CAD in breast imaging have given way to the realization that traditional CAD may yield limited or no increases in diagnostic performance [4]. Multiple recent studies have raised concerns about the cost-effectiveness and clinical utility of CAD in breast imaging. An observational study of community-based mammography facilities from the Breast Cancer Surveillance Consortium performed by Fenton et al. between 1998 and 2002 found that CAD use reduced overall radiologist reading accuracy as evaluated by receiver operating characteristic curve analysis [5]. A subsequent study published in July 2011 found that CAD use reduced specificity by increasing recall rates, with no increase in sensitivity or invasive tumor characteristics (stage, size, or lymph node status) [6]. Despite these concerns about its efficacy, CAD utilization for screening mammograms has become ubiquitous, with reimbursement bundled with screening mammography, and with utilization in 92% of all screening mammograms performed in the United States in 2016 [7]. The failure of conventional CAD to fulfill the need for improving and optimizing mammography performance creates a continued opportunity for artificial intelligence (AI) in breast imaging.

## 3. Artificial Intelligence

AI is a large field that includes many diverse technologies and applications with the shared characteristics of using computer-based algorithms and data to solve problems or perform tasks that would typically require human intelligence. In the past 10–15 years, there have been tremendous advances in the availability and accessibility of more powerful computational hardware for processing and storing data needed for AI applications. At the same time, and perhaps even more critically, there has been an exponential increase in the amount and availability of data for training AI algorithms. These changes have allowed for revolutionary developments in AI during the past 10 years, with particular focus on machine learning (ML). ML is a subset of AI in which computers are trained and perform functions without being explicitly programmed by humans on how to complete those tasks. ML commonly uses features and input from human programmers as the basis of learning. Further along the continuum of ML is representation learning, which does not require human feature engineering, but rather involves a system learning the features itself. Deep learning (DL) is a step further, where the features are extracted in a hierarchical fashion and with many simple features making up more complex features [8]. These changes and developments have allowed for DL applications that generate truly groundbreaking performance enhancements in image analysis tasks [8].

DL utilizing convolutional neural networks has seen an explosion of possibilities and practical uses for image analysis for non-medical images in the past 10 years. This includes many non-medical imaging related tasks such as image classification or detection, which are already deeply ingrained in daily workflows. These successes have led to interest for applications within radiology that could apply the success of AI algorithms in image analysis to perform clinically meaningful tasks such as classification (presence or absence of disease), segmentation (quantitative analysis of organs or lesions for surgical planning), and detection (determining the presence or absence of a lesion or nodule) amongst many other diverse applications for AI in radiology [8].

## 4. Artificial Intelligence in Breast Imaging

### 4.1. Opportunities in Breast Imaging for AI Applications

Breast imaging has many unique features and characteristics that create opportunities for meaningful AI applications (Table 1). Specifically, the longstanding and unique structured lexicon of breast imaging as defined by the Breast Imaging Reporting & Data System (BI-RADS^®^) from the American College of Radiology facilitates the development and implementation of AI. BI-RADS^®^ provides a standardized and structured system of lexicon and terminology, reporting, classification, communication and medical auditing for mammography, breast ultrasound, and breast MRI [9]. This system supports the development and evaluation of AI applications in breast imaging in many ways, perhaps most importantly by creating a predefined methodology and framework for the radiologist’s interpretation of breast imaging studies and the mapping of results. When combined with medical outcomes, auditing, and reporting, there is a repository of data for breast imaging included in radiologist interpretations and clinical outcomes for mammography [10]. Moreover, the standardized approach to screening mammography where two specific mammographic positions are imaged for each breast (craniocaudal and mediolateral oblique positions) improves the standardization of imaging data being utilized for training and validation.

This standardization and established methodology for determining and tracking results has facilitated the creation of multiple large data sets which are a prerequisite for the development of high-performing AI algorithms. There are currently multiple large mammography data sets, some of which contain more than 1 million mammograms with associated patient factors and known clinical outcomes [11,12,13,14]. Many of the available data sets come from various sources including different practice locations, practice types, and multiple mammography vendors. Some data sets are also focused on including a racially diverse case mix, which is critical to ensuring high levels of performance across the entire population [11]. The availability of data sets is significantly more advanced for mammography, in particular screening mammography, when compared to other breast imaging modalities such as ultrasound or MRI.

### 4.2. Challenges of Breast Imaging for AI Applications

There are several unique aspects of breast imaging that make the development and implementation of high-performing AI algorithms more challenging (Table 1). For example, the recent rapid adoption and widespread utilization of digital breast tomosynthesis (DBT) has created challenges on multiple fronts. The image data for DBT are unique and significantly different from standard full-field digital mammography (FFDM), as many slices of images are created with each mammographic position versus the single FFDM image for each view. The appearances of the images, including benign and malignant pathology, differ significantly. Moreover, DBT file sizes are orders of magnitude greater than those of traditional FFDM images, with files sizes for single exams approaching or exceeding 1 gigabyte. This creates significant challenges for the storage, transfer, and consumption of this large volume of data, particularly in a busy clinical application.

There are also significant variations in the appearance of DBT images between various vendors, with the differences being significantly greater than when comparing traditional FFDM mammographic images. Further compounding these challenges are the recent but variable use of synthetic mammographic images to replace traditional FFDM images. These synthetic images are generated from tomosynthesis imaging data, a method which has the advantage of eliminating the need for standalone FFDM images and thus significantly reduces the radiation dose for patients. The utilization of synthetic mammography is highly variable [15]. Moreover, there are significant differences between the appearances of these synthetic images between vendors and between software upgrades of a vendor. This variability, and the recent heterogenous adoption of these technologies, has created a significant limitation and challenge for AI algorithms which may not have been developed for a certain image type or may not perform equally well across all vendors and systems.

An additional significant challenge for the development of high-performing AI algorithms is the manner in which breast images are interpreted. Frequently, breast imaging studies are interpreted with the utilization of multiple prior comparison imaging exams, allowing radiologists to identify new, subtle, meaningful changes and dismiss stable, benign variations. This is an additional process that must be either built into AI algorithms or otherwise accounted for in their application. Additionally, the interpretation of breast imaging is commonly a multimodal process, particularly outside of the screening mammography environment. Often, mammography is used in conjunction with breast ultrasound, breast MRI, or other adjunctive imaging modalities to evaluate and work up breast problems. These tasks are also performed against a complex background in which the radiologist aggregates information in real time about patients’ clinical and medical history, including risk scores amongst other factors, referring providers, and technologists, which can influence the most efficacious workup and diagnosis. These different, disparate data sources and factors represent challenges for the development of high-performing AI algorithms.

## 5. Applications for Artificial Intelligence in Breast Imaging

Breast cancer screening programs rely on screening mammography to reduce the morbidity and mortality of breast cancer. Much of the published literature and available AI applications focus on cancer detection for mammography. However, in addition to cancer detection, there are several potential interpretive and non-interpretive applications for AI in breast imaging including decision support, risk assessment, breast density quantitation, workflow and triage, quality evaluation, response to neoadjuvant chemotherapy assessment, and image enhancement (Table 2).

## 6. Cancer Detection

Much of the research, development, and excitement surrounding AI applications in breast imaging have been focused on cancer detection, notably cancer detection in screening mammography (Table 3). The widespread adoption of breast cancer screening programs, the significant morbidity and mortality of breast cancer worldwide, and the efficacy of high-performing screening mammography create a unique and powerful opportunity for AI. This has led to a tremendous focus on AI-based applications for mammography-based breast cancer detection.

There have been many published examples of AI algorithms which demonstrate excellent performance in cancer detection for screening mammography. These include a number of algorithms trained and evaluated on single-institution or homogenous internal data sets. However, there have also been multiple, more recent examples of AI algorithms trained on larger, more heterogenous or representative data sets. These includes an AI-based cancer detection system trained on United Kingdom (U.K.) and United States (U.S.) data comparing AI performance vs. radiologist performance in a reader study finding absolute reductions of 5.7% and 1.2% in false positives and 9.4% and 2.7% in false negatives (U.S. and U.K. data sets, respectively) [16]. The AI algorithm performed significantly better than all human readers in the reader study [16]. Another seminal AI algorithm was developed as the result of an international crowdsourcing challenge which found that individual AI algorithms approached but did not exceed radiologist performance [17]. Rather, the best performance was achieved when an ensemble of the best AI algorithms was combined with a radiologist [17]. An additional published AI model trained on more than 1 million images achieved an excellent AUC for cancer detection of 0.895 when evaluated on a large data set. Further evaluation and comparison of this model’s performance with a group of radiologists in a reader study found that the AI’s AUC exceeded that of all individual readers; however, it was importantly found that the performance of a radiologist–AI hybrid model was the highest in the reader study, exceeding both the individual and AI-alone performances [18].

Such investigations have generated tremendous excitement for using AI applications for breast cancer detection, with multiple commercially available products already available for use on the market, in addition to other investigational or open-source AI algorithms. However, there is currently a significant gap in the understanding of how these AI applications will perform in the real world when used in clinical practice by radiologists. A recent review article found no prospective studies testing accuracy of AI in screening practice [19]. The review also noted significant issues with methodology and quality in published investigations, finding the majority of AI applications were less accurate than a single radiologist and that all included algorithms were less accurate than a consensus of two or more radiologists. The authors also noted the pattern of small, more limited studies finding AI to be more accurate than radiologists demonstrated issues with bias and generalizability, and that their results were not yet replicated in larger studies [19].

External attempts to evaluate the performance of AI algorithms have resulted in variable observed performance. For example, a high-performing AI model demonstrated significantly inferior performance when used at an external site in its native form [20]. This same AI model was then tested after training without transfer learning and after retraining with transfer learning (using a pretrained algorithm and then applying it to a new but related problem with some modification). Local retraining of that model with transfer learning allowed for improvement in performance that approached the initially reported levels [20]. The results suggest limitations and concerns regarding generalizability of performance in AI applications. Perhaps more importantly, these results illustrate the possibilities for optimizing AI performance locally at sites using a generally available model. A study looking to externally evaluate and compare three different commercially available algorithms with human readers (single and double) found that performance for the single best AI algorithm was sufficiently high that it could be evaluated as an independent reader for screening mammography [21]. Moreover, combining the first radiologist reader with the best AI algorithm found more cancers than using the first and second radiologist readers. A systematic review of independent external validation of AI algorithms for cancer detection in mammography found only 13 data sets that met the inclusion criteria, with all being either retrospective reader or simulation studies. The review found mixed results, with only some AI algorithms alone exceeding radiologist performance, whereas in all reviewed instances radiologists combined with AI outperformed radiologists alone [22]. An additional serious concern is that developed and available AI algorithms may not perform equally well in all subpopulations or patient groups. A study evaluating a well-known, previously externally validated, high-performing AI algorithm on an independent, external, diverse population found that for certain patient groups, it had much lower performance compared to other groups and previously published performances. These issues raise concerns about unintended secondary consequences of inadequate inclusion of all patient groups in testing and validation data sets [23].

DBT represents an additional challenge when interpreting the available literature and evaluating AI performance for clinical use. Many of the large available data sets for training and previously published AI algorithms were created and validated entirely or predominantly using FFDM data sets. As DBT has gained widespread adoption and a high level of utilization, this creates more uncertainty when attempting to generalize the expected performance of commercially available or investigational AI algorithms into clinical practice. A recent, large study evaluating a well-known commercially available AI algorithm’s performance on FFDM versus DBT found significantly diminished levels of performance for the AI algorithm with the DBT data [24]. The AI algorithm met or surpassed radiologist performance for FFDM but generated a markedly higher and undesirable false-positive rate with DBT images, illustrating both the challenges of these data and the difficulty in generalizing performance across settings [24].

The high level of interest and focus on developing improved AI algorithms for screening mammography has led to the RSNA Screening Mammography Breast Cancer Detection Competition of 2023 [25]. This competition will utilize data from the ADMANI data set and should further increase the attention and resources dedicated to breast cancer detection with AI applications [14].

The development and evaluation of AI tools for cancer detection in breast imaging have overwhelmingly been focused on mammography. This is intuitive given the immense number of mammography exams that are performed worldwide and the relatively standardized nature of mammography. There have however been additional investigations regarding AI to increase cancer detection with breast ultrasound, breast MRI, and contrast-enhanced mammography. A recent retrospective reader study evaluating hybrid AI and radiologists’ performance in the interpretation of screening and diagnostic breast ultrasounds found preserved sensitivity for breast cancer detection with the hybrid AI workflow, with the advantage of reducing false positives by 37.3% and decreasing benign biopsies by 27.8% for screening ultrasounds [26]. Screening breast ultrasounds can be performed using a handheld technique or an automated technique. There are several benefits of automated breast ultrasound technology for screening; however, such studies typically contain significantly more than 1000 images, which presents a significant challenge for radiologists’ efficiency and can lead to failure to detect a meaningful finding. A commercially available AI-powered application is available for use as CAD in automated breast ultrasound studies which may be able to help address these challenges for automated breast ultrasound screening [27]. AI applications for cancer detection with breast MRI are also in development, including a recent study reporting non-inferiority between breast radiologists and an AI system for identifying malignancy in breast MRI [28]. Contrast-enhanced mammography is another important supplemental screening modality whose use is evolving rapidly. A DL model developed to evaluate contrast-enhanced mammography images demonstrated excellent performance, and radiologist use of the AI model led to significantly improved performance metrics for radiologists in the study [29].

## 7. Decision Support

Improving the efficacy of breast imaging interpretations is not restricted to cancer detection on screening exams. There is also a need to improve radiologists’ diagnostic performance when a lesion or area of interest is identified. Many opportunities for improvement are available in the realm of decision support, including limiting benign biopsy recommendations and minimizing false-negative interpretations. Applications of decision support have been studied in several different scenarios across various breast imaging modalities. A study evaluating an AI-based clinical decision support application for DBT found that radiologists using the decision support system were able to increase sensitivity while preserving specificity, thus reducing the likelihood of false-negative interpretations without increasing benign biopsy recommendations [30]. A separate investigation evaluating AI decision support for mammography evaluated radiologist performance in categorizing masses, finding improved AUC when using AI decision support with both increased sensitivity and specificity [31]. The authors also found that more junior radiologists made more interpretive adjustments for masses that were suspicious when using AI decision support, suggesting experience or confidence may be an important potential variable for the impact of decision support. Another study evaluating an AI-based algorithm used images and clinical factors for predicting malignancy among suspicious microcalcifications seen on mammography, a common diagnostic problem encountered in breast imaging, demonstrating non-inferior diagnostic performance compared to a senior breast radiologist and outperforming junior radiologists [32].

Decision support opportunities in breast imaging extend beyond mammography. A common diagnostic problem encountered is appropriately stratifying breast masses identified on an ultrasound as either benign, needing short-term follow-up, or requiring biopsy for tissue diagnosis. A multicenter retrospective review of a commercially available AI breast ultrasound decision support application found that radiologist reader performance increased significantly when using the AI decision support system, with the AUC increasing from 0.83 without decision support to 0.87 with decision support [33]. Interestingly, the same study found that using decision support can reduce intrareader variability, providing an opportunity to standardize interpretive performance [33]. In a recent retrospective study evaluating decision support for breast MRI, radiologist readers from academic and private-practice centers compared radiologists reading with conventional MRI CAD software versus AI-based MRI CAD software. The study found the AUC significantly improved with the AI algorithm for all readers, with an average improvement from 0.71 to 0.76 [34]. These findings suggest a role for improving diagnostic performance within the context of complex breast MRI interpretations.

## 8. Breast Density

Breast density reflects the mammographic amount of fibroglandular tissue in the breast, designated into four categories by BI-RADS: (a) almost entirely fatty, (b) scattered areas of fibroglandular density, (c) heterogeneously dense, which may obscure small masses, and (d) extremely dense, which lowers the sensitivity of mammography. Approximately 40% of women in the United States have dense breasts, designated as category c or d [35]. Breast density is an independent risk factor for breast cancer, with at least a moderate association with breast cancer risk [36]. Due to this elevated risk and the decreased sensitivity of mammography for them, women with dense breasts may benefit from supplemental screening with modalities such as breast ultrasound, contrast-enhanced mammography, molecular breast imaging, or breast MRI. In most states across the United States, women are required by law to be notified of their breast density. Recently, the FDA issued a national requirement for breast density notification, which will go into effect in September 2024 [37].

The accuracy of breast density reporting can be subject to interpersonal and intrapersonal variability amongst radiologists, highlighting the value of computer-based assessment in the standardization of breast density reporting. Early iterations required manual input to outline and define breast tissue density [38]. Numerous fully automated DL algorithms are now available which use convolutional neural networks to define breast density, demonstrating high levels of accuracy in stratifying dense and non-dense breasts. For example, an externally validated algorithm demonstrated 89% accuracy in stratifying non-dense and dense breasts, with a 90% agreement between the algorithm and three independent readers [39]. Other models have also demonstrated high levels of agreement in clinical use in the binary categorization of dense and non-dense breasts, with 94% agreement amongst radiologists with a DL algorithm when evaluating more than 10,000 mammography examinations [40]. Diagnostic accuracy can be maintained in algorithms assessing breast density in synthetic mammograms, demonstrating an accuracy of 89.6% when differentiating dense and non-dense breasts [41]. However, the possibility for altered performance of automated breast density assessments exists when moving from FFDM images to synthetic mammography, including complex potential interactions with ethnicity and body mass index that require awareness and attention [42]. Numerous FDA-approved algorithms for quantification of breast density are currently available for use, including some widely used in clinical practice (Table 2). A study comparing mammographic density assessment in these models demonstrated that the percentage density measured by some specific commercially available algorithms also had a strong association with breast cancer risk, suggesting there may be utility in automated density assessment in cancer risk stratification [43].

## 9. Cancer Risk Assessment

Identifying women at increased risk of breast cancer is as an important assessment when determining the need for additional screening and preventative intervention. Current risk assessment models estimate the average risk of breast cancer for women with similar risk factors, as opposed to individual breast cancer risk. These models include the Gail model (BCRAT), Tyrer–Cuzick model (IBIS), Breast and Ovarian Analysis of Disease Incidence and Carrier Estimation Algorithm model (BOADICEA), BRCAPRO, and Breast Cancer Surveillance Consortium model (BCSC). Each of these models accounts for different factors such as age, age of menarche, obstetric history, first-degree and multi-generational relatives with breast cancer, genetic information, number of previous biopsies, race and ethnicity, and body mass index, amongst other factors. The models calculate 5-year, 10-year, or lifetime risk of breast cancer and are used to identify women who may benefit from supplemental high-risk screening for breast cancer, chemoprevention, or lifestyle modifications. As the different models rely on unique combinations of risk factors, including some factors while excluding others, there are several limitations to a sole model being used to independently predict cancer risk. For example, the Gail model can underestimate the risk of breast cancer in women with familial history of breast cancer or a personal history of atypia, as well as in non-American and non-European populations [44]. In a study evaluating the 10-year performance of the Gail, Tyrer–Cuzick, BOADICEA, and BRCAPRO models, the authors identified that the integration of multigenerational family history, such as in the Tyrer–Cuzick and BOADICEA models, better demonstrates the ability to predict breast cancer risk [45]. This analysis also suggested that a hybrid model incorporating various factors from each of these models may help improve accuracy in breast cancer detection risk. A separate cohort analysis comparing these 5 risk assessment models in 35,000 women over 6 years demonstrated similar, moderate predicative accuracy and good overall calibration amongst the models (AUC 0.61–0.64) [46].

New developments in AI image-based risk models demonstrate promising results in cancer risk assessment, in some instances outperforming traditional cancer risk assessment models. A case-cohort study of an AI image-based mammography risk model assessed the short-term and long-term performance of its model compared to the Tyrer–Cuzick version 8 model over a period of 10 years [47]. The image-based AI model outperformed the Tyrer–Cuzick model in both short-term and long-term assessment when evaluating approximately 8600 women, with age-adjusted AUC AI model performances ranging from 0.74 to 0.65 for breast cancers developed in 1 to 10 years, significantly exceeding the Tyrer–Cuzick age-adjusted AUCs of 0.62 to 0.60 in this time period [47].

Mirai, a DL mammography-based risk model, incorporates digital mammographic features along with clinical factor inputs to provide breast cancer risk prediction within 5 years and was recently validated across a broad, diverse international data set [48]. Approximately 128,000 screening mammograms and pathologically confirmed breast cancers across 7 international sites in 5 countries including the United States, Israel, Sweden, Taiwan, and Brazil, were evaluated [48]. Of the 62,185 unique patients, 3815 patients were diagnosed with breast cancer within 5 years of the index screening mammogram, with Mirai obtaining concordance indices of >/=0.75 and AUC performances of 0.75 for White women (0.71–0.78, 95% CI) and 0.78 for Black women (0.75–0.82, 95% CI), outperforming traditional cancer risk models. Such a model demonstrates the promise of an AI cancer risk assessment tool to significantly improve the accuracy of breast cancer risk assessments. Moreover, making personalized AI image-based assessments is an opportunity for improved performance for all ethnicities and groups, including those for whom previous risk assessment models did not perform as well.

## 10. Workflow Applications

AI-based triage tools can be used to prioritize patients and improve overall workflow for radiologists interpreting breast imaging studies. This has been most well-studied with screening mammography. Using AI-based triage algorithms, a retrospective simulation study in which AI-based screening (normal—no radiologist, moderate risk—radiologist review, and suspicious—recalled) was compared to radiologist screening found non-inferior sensitivity and higher specificity (with a 25.1% reduction in false positives) [49]. The findings of this study were achieved while simultaneously achieving a workload reduction of 62.6%, with triaged normal studies read only by the AI system. Similarly, another retrospective simulation study showed that using AI to triage mammograms into no-radiologist assessment and enhanced assessment categories could potentially reduce workloads by more than 50% and preemptively detect a substantial proportion of cancers otherwise diagnosed later [50]. These findings suggest a novel potential way of integrating AI-based cancer detection into clinical workflows to preserve or improve clinical performance while reducing workloads. The implications for this type of workflow may differ between screening programs with a single- versus a double-reader paradigm.

Another study evaluating an AI system used in the detection of lesions on DBT found that when the algorithm was concurrently incorporated into the interpretation of the mammograms, it reduced reading times by approximately half while still improving accuracy with a statistically significant 0.057 average improvement in AUC [51]. As reading times with DBT are significantly longer than with FFDM, this provides an opportunity for increased efficiency, which is particularly important given the current shortage of trained radiologists who can interpret mammograms. An alternative approach for improving interpretive efficiency for DBT is the replacement of traditional 1 mm thin tomosynthesis slices with 6 mm thick overlapping slices that has been implemented by a mainstream mammography modality manufacturer [52]. These thick slices are created in part by using AI algorithms to make salient suspicious findings more conspicuous [52]. This should allow for increased efficiency in the interpretation of DBT by reducing the number of slices for review. By using these triage tools, radiology practices could prioritize examinations to be read immediately, categorize cases by complexity, and replace the second reader at sites, offering double reading to enhance radiologist workflows [53]. As of today, there are multiple commercially available algorithms that can assist in triage of mammographic interpretation (Table 2).

## 11. Quality Assessment

The importance of maintaining high-quality positioning and technique has long been a focus for mammography. MQSA includes a significant focus on ensuring standardization and quality for mammography in the United States. Poor positioning is often identified as a leading cause of clinical imaging deficiencies and misdiagnosis [54]. This has led to the recently implemented FDA EQUIP initiative that began in 2017 to emphasize and focus on ensuring and improving quality for the effective performance of mammography. The need for uniform, high-quality mammographic technique and positioning creates an opportunity for AI algorithms to evaluate mammography exams and provide feedback and opportunities for improvement for performing technologists and interpreting physicians. A recent study found an AI algorithm could assess breast positioning on mammography to search for common issues that can lead to inadequate positioning, such as nipple in profile, breast rotation, visualization of the pectoral muscle, inframammary fold, and the pectoral nipple line, with the algorithm being highly accurate in identifying these deficiencies [55]. Additional research studying the application of AI to breast positioning assessment has looked to replicate additional quality assessment tasks performed by radiologists when interpreting mammograms in hopes of standardizing the detection of these issues, finding some success as well [56]. In fact, there is currently a commercially available application that utilizes AI to help evaluate, track, and improve quality in mammographic positioning [57].

## 12. Neoadjuvant Chemotherapy Response

AI may also be used to assess treatment response to neoadjuvant chemotherapy for breast cancer. Neoadjuvant chemotherapy (chemotherapy given prior to surgery) can reduce tumor size, allowing for less-invasive surgical procedures. It also enables in vivo evaluation of treatment response, allowing therapeutic treatment plans to be modified based on each patient’s individual response [58]. Despite its relatively low sensitivity (63–88%) and specificity (54–91%), MRI is currently the most accurate imaging method for determining tumor response to neoadjuvant therapy [59]. Recent research has demonstrated that AI has the potential to improve treatment response prediction. A meta-analysis by Liang et al. found that ML and MRI are highly accurate (AUC = 0.87, 95% CI = 0.84 to 0.91) in predicting responses to neoadjuvant therapy [60].

AI applied to imaging may predict tumor response to treatment prior to the *initiation* of neoadjuvant chemotherapy. A proof-of-concept study by Skarping et al. demonstrated the effectiveness of a DL-based model using baseline digital mammograms to predict patient responses to neoadjuvant therapy, with an AUC of 0.71 [61]. Their model predicted tumor response by deciphering breast parenchymal patterns and tumor appearances as reflected by different grey-level pixel presentations in digital mammography. This type of platform may help aid in clinical decision-making prior to administering chemotherapy, significantly reducing patient morbidity. Likewise, a study evaluating ultrasound images of primary breast cancer in clinically node-negative patients was able to predict the likelihood of having lymph node metastases at surgery with a high level of accuracy [62]. These findings demonstrate the evolving possibilities for AI-based applications to positively predict patient outcomes and may provide opportunities to individually tailor and improve patient care.

## 13. Image Enhancement

There have been several recent novel investigations and developments using AI algorithms to enhance the appearance of images in breast imaging. One creative example is the use of an AI-based process that first involved collapsing or merging suspicious regions of interest from DBT into ‘maximum suspicion projections’ that emphasize the suspicious findings, making them more conspicuous [63]. These novel synthesized images are then used as an input for an AI cancer detection model to detect breast cancers, reducing the burden of image and data preparation. Along the same lines of this approach, a major mammography modality vendor now has a commercially available AI-based application that emphasizes features that are likely to be important for accurate imaging review, such as bright foci which may represent calcifications, lines that can represent distortion, or rounded objects that may be masses [52]. This information is derived from 1 mm slices from the DBT images but is then combined into overlapping, thick 6 mm slabs that can significantly decrease the number of slices that need to be reviewed while still preserving the visibility of salient findings [52].

Additional investigations have evaluated using AI applications to reduce the amount of intravenous contrast dose needed for breast MRI examinations [64]. This is especially important given the current recommendations for serial annual supplemental screening breast MRIs for patients at high risk for breast cancer and the recent focus on the possibility of gadolinium retention. A recent study demonstrated an AI algorithm that was developed using a data set of breast MRI images with and without contrast. This AI model was then given inputs consisting only of non-contrast images from breast MRI studies and was able to generate simulated contrast-enhanced breast MRI images [65]. These simulated images were felt to be quantitatively similar to, and demonstrated high level of tumor overlap with, the true contrast-enhanced breast MRI images, with 95% of images found to be of diagnostic quality by the study radiologists. These developments demonstrate the power of AI applications to create clinical value and novel potential workflows using minimal or limited data sets.

## 14. Discussion and Future Directions

There are at least 20 available FDA-approved AI-based applications available today for breast imaging (Table 2). Beyond these currently commercially available applications, there are many more areas for AI in breast imaging that are being investigated and developed. These potential areas for AI applications to impact breast imaging are at various degrees of maturity and availability (Figure 1).

However, there are significant barriers to the implementation of AI applications in breast imaging, including inconsistent performance, significant cost, and IT requirements, along with the lack of radiologist, patient, and referring provider familiarity and trust [66]. Additionally, there are meaningful concerns for the generalizability of AI algorithms in breast imaging, with a recent publication showing significant performance degradation of an AI algorithm that was trained using images from a specific manufacturer when tested using an updated system/software from that same manufacturer [67]. This required site-specific modification of the algorithm to improve its performance. These issues demonstrate a more general concern for the ability of AI applications to generate consistent and uniform results between sites and clinical scenarios. This reinforces the need for careful evaluation of applications for each site and close monitoring of performance. Further complicating adoption is the lack of reimbursement for AI applications in breast imaging, which may drive focus and adoption towards applications that can provide convincing workflow or efficiency gains to counterbalance the costs of adoption and implementation. Additional obstacles to the successful use of AI in breast imaging include a lack of understanding of how radiologists interact with AI applications. This includes concerns about how biases may lead inexperienced radiologists to over rely on AI applications, resulting in diminished clinical performance [68].

During the first few months of 2023, there has been tremendous excitement and focus on a handful of AI natural language processing models, specifically large language models, that seem poised to generate evolutionary and disruptive change throughout many different fields and industries. ChatGPT, a conversation large language model, is perhaps the most well-known and discussed of these models and is extremely successful at automatically summarizing large inputs of information and answering questions in a conversational manner. Potential applications within breast imaging may include imaging appropriateness and clinical decision support, preauthorization needs, generating reports, summarizing information from electronic medical records, and creating interactive computer-aided detection applications [69]. Given the high degree of contact of breast imaging with patients and general population applications like ChatGPT, large language models may provide value in shaping and guiding patient interaction and education for breast imaging topics in the future.

With the expanding use of AI in breast imaging, it will be necessary in the future to define the roles of radiologists and AI applications in clinical care to maximize the clinical benefit. These roles may change over time but in the near future may include tasks that are best served by AI alone, radiologists alone, or AI and radiologists together (Figure 2).

In addition to defining the roles that AI and radiologists will play in patient care, it will be essential to build and shape the trust and perceptions of patients and referring providers towards AI in breast imaging. Patient attitudes and perceptions regarding AI in radiology are complex and include matters of distrust and accountability, concerns about procedural knowledge, a preference for preserving personal interaction, efficiency, and remaining informed about use [70]. More generally, approximately 50% of women of screening age (over 50) in England report positive feelings about the use of AI in reading mammograms, with the remainder being neutral or reporting negative feelings [71]. These data suggest that there will be significant future work towards educating patients on how AI can be implemented in breast imaging and keeping patients aware of the benefits and limitations of its use.

## 15. Conclusions

The rapid evolution and expansion of the use of AI in breast imaging presents a tremendous opportunity to improve the quality of breast imaging provided for patients. Historically, the biggest challenges were the availability of sufficient computational power and data sets for creating and training AI applications. Today, some of the most significant challenges are validating AI performance across diverse data sets and patient populations and overcoming implementation barriers. In the future, the key challenges will likely include better defining the roles that AI should play in patient care and then communicating these decisions effectively to referring clinicians and patients. The value and opportunity of AI in breast imaging are clear, and as these challenges are met in the future, the full potential of AI to transform breast imaging can be realized.

## Figures and Tables

**Figure 1 diagnostics-13-02041-f001:**
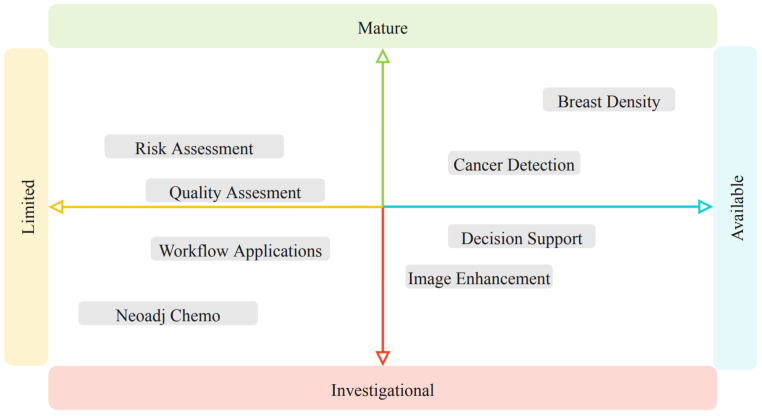
Visual representation of maturity and availability of different areas for AI applications for breast imaging.

**Figure 2 diagnostics-13-02041-f002:**
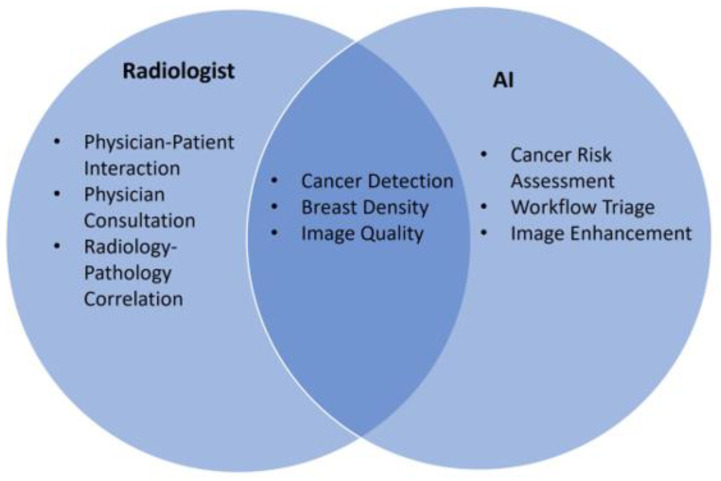
Venn diagram demonstrating possible future situations where radiologists or AI can make contributions separately and together towards breast imaging care.

**Table 1 diagnostics-13-02041-t001:** Summary of Opportunities and Challenges for AI in Breast Imaging.

Breast Imaging Opportunities	Breast Imaging Challenges
Longstanding uniform reporting and lexicon	Recent widespread adoption of digital breast tomosynthesis (DBT)
Mandated robust outcomes and clinical results tracking systems	Variability in image appearance between vendors, increasing with DBT and synthetic 2D mammography
Standardized positioning and technique	File sizes are extremely large
Large available data sets for training	Breast imaging interpretation can rely on concurrently performed mixed-modality (mammography, ultrasound, MRI) studies
Familiarity with and acceptance of computer-aided detection (CAD)	Clinical information obtained from patient, referring provider, and technologists key for accurate interpretation

**Table 2 diagnostics-13-02041-t002:** Breast Imaging Artificial Intelligence Potential Applications.

Interpretive AI	Non-Interpretive AI
Cancer Detection	Cancer Risk Assessment
Decision Support	Density Quantification
Response to Neoadjuvant Therapy	Workflow Triage
	Image Enhancement
	Image Quality Assessment

**Table 3 diagnostics-13-02041-t003:** Summary of FDA-approved AI Applications in Breast Imaging.

Product Name	Vendor	Country of Origin	Modality
**Cancer Detection**			
cmAssist^®^	CureMetrix	United States	Mammography
Genius AI™ Detection	Hologic^®^, Inc.	United States	Mammography and Tomosynthesis
Lunit INSIGHT MMG	Lunit	South Korea	Mammography
MammoScreen^®^ 2.0	Therapixel	France	Mammography and Tomosynthesis
ProFound AI^®^	iCAD, Inc.	United States	Mammography and Tomosynthesis
Saige-Dx™	DeepHealth, Inc.	United States	Mammography
Transpara^®^	ScreenPoint Medical B.V.	Netherlands	Mammography and Tomosynthesis
**Decision Support**			
Koios DS™ Breast	Koios™ Medical, Inc.	United States	Ultrasound
QuantX™	Qlarity Imaging	United States	MRI
**Density Quantification**			
cmDensity™	CureMetrix, Inc.	United States	Mammography
IntelliMammo^™^ densityai™	Densitas^®^	Canada	Mammography
PowerLook^®^ Density Assessment	iCAD, Inc.	United States	Mammography
Quantra™ 2.2	Hologic^®^, Inc.	United States	Mammography and Tomosynthesis
Saige-Density™	DeepHealth, Inc.	United States	Mammography and Tomosynthesis
Syngo.BreastCare	Siemens^®^	Germany	Mammography
Visage Breast Density	Visage Imaging, Inc.^®^	United States	Mammography
Volpara TruDensity^®^	Volpara Imaging	New Zealand	Mammography
WRDensity	Whiterabbit.ai	United States	Mammography
**Triage**			
cmTriage^®^	CureMetrix, Inc.	United States	Mammography
HealthMammo	Zebra Medical Vision	Israel	Mammography
Saige-Q™	DeepHealth, Inc.	United States	Mammography and Tomosynthesis
Syngo.BreastCare	Siemens^®^	Germany	Mammography

## Data Availability

Not applicable.

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
