# Peer review of "Artificial Intelligence Applications in Breast Imaging: Current Status and Future Directions"

_diagnostics, 2023, doi:10.3390/diagnostics13122041_

Round 1

Reviewer 1 Report

The paper needs more organization, the current usage of AI in breast cancer diagnosis needs more clarification, the conclusion part is missing, and the paper should be provided with graphs and figures that make the text more readable. The authors should discuss the obstacles of AI in breast cancer diagnosis. There are other modalities that should be mentioned and what are the limitations for each one? Why AI should be supported by these modalities. 

The future of AI in breast diagnosis should be more clarified.

I suggest that authors add tables to summarize their paper and tables to compare the most significant issues in the past, current, and future of AI in breast diagnosis. 

Author Response

We appreciate the thoughtful response and suggestions found in this review to improve the manuscript. We have carefully considered the recommendations and made significant changes to address each recommendation. These are listed below.

The paper needs more organization, the current usage of AI in breast cancer diagnosis needs more clarification, the conclusion part is missing, and the paper should be provided with graphs and figures that make the text more readable.  

  1. We have made the recommended modifications regarding organization of the discussion of the different types of applications of AI in breast imaging (lines 174-180). We also added Table 2 as a visual representation to help the reader organize and better understand the remainder of the paper.
  2. We have added additional tables and figures to help make the paper more easily readable. See Figure 2 and Table 2.
  3. We have added a conclusion section. 

The authors should discuss the obstacles of AI in breast cancer diagnosis.  

  1. The obstacles to breast cancer diagnosis are discussed in the Cancer Detection section, included in lines 218-230 and lines 252-263. Some of these obstacles include accuracy when compared to radiologists, challenges in external validation, and variability in image appearance between vendors, particularly with widespread adoption of DBT.  
  2. We have also expanded the discussion on obstacles to AI in the discussion section and added an additional very recent citation related to bias in breast imaging related to AI applications. 

There are other modalities that should be mentioned and what are the limitations for each one? Why AI should be supported by these modalities.  

  1. As a review article we limited our discussion to AI applications that are based on breast imaging modalities that are in widespread clinical use. We included specific discussion for mammography, ultrasound, MRI, and contrast enhanced mammography which constitute the current recommendations for breast imaging for society/national guidelines. 

The future of AI in breast diagnosis should be more clarified. 

  1. Throughout the paper there are multiple descriptions of the current state and future directions for AI applications in Breast Imaging. This is also specifically addressed in the discussion and conclusion sections. 

I suggest that authors add tables to summarize their paper and tables to compare the most significant issues in the past, current, and future of AI in breast diagnosis. 

  1. We have specifically addressed this in the new conclusion section.
  2. We have added lines 174-180 to summarize potential applications of artificial intelligence in breast imaging as detailed in the manuscript, divided into interpretive and non-interpretive AI and included new table 2. 

Reviewer 2 Report

1) It is necessary to supplement the review with the technology of diagnosing breast cancer based on the measurement of temperature fields using microwave radiometry. This diagnostic is based on various artificial intelligence algorithms. This will give the reader a better understanding of the problems of cancer diagnosis.

2) The meaning of figure 1 is not clear. We need to make it more saturated and specific.

Author Response

We appreciate the thoughtful response and suggestions found in this review to improve the manuscript. We have carefully considered the recommendations and made significant changes to address each recommendation. These are listed below.

It is necessary to supplement the review with the technology of diagnosing breast cancer based on the measurement of temperature fields using microwave radiometry. This diagnostic is based on various artificial intelligence algorithms. This will give the reader a better understanding of the problems of cancer diagnosis. 

  1. As a review article we limited our discussion to AI applications that are based on breast imaging modalities that are in widespread clinical use. As microwave radiometry for breast imaging is not currently widely used in clinical practice and is not addressed or included in any specific society/national guideline we did include a discussion of AI based applications that use microwave radiometry.  

2) The meaning of figure 1 is not clear. We need to make it more saturated and specific. 

  1. Figure 1 is intended as a visual representation of the current status of AI applications in breast imaging with some types of applications being more well studied or more widely available than others. Additional description and introduction are added to specify the contents of the figure including in the body of the text an din the figure legend. 

Round 2

Reviewer 1 Report

I think the authors considered my comments